# Broken Balance: Emerging Cross-Talk Between Proteostasis and Lipostasis in Neurodegenerative Diseases

**DOI:** 10.3390/cells14110845

**Published:** 2025-06-04

**Authors:** Jessica Tittelmeier, Carmen Nussbaum-Krammer

**Affiliations:** Chair of Neuroanatomy, Institute of Anatomy, Faculty of Medicine, Ludwig-Maximilians University of Munich (LMU Munich), Pettenkoferstrasse 11, 80336 Munich, Germany

**Keywords:** neurodegenerative diseases, prion-like propagation, lysosome, lysosomal lipid storage diseases, sphingolipidoses, proteostasis, lipostasis

## Abstract

Neurodegenerative diseases, including Alzheimer’s disease and Parkinson’s disease, are characterized by progressive neuronal loss, leading to cognitive and motor impairments. Although these diseases have distinct clinical manifestations, they share pathological hallmarks such as protein aggregation and lysosomal dysfunction. The lysosome plays a vital role in maintaining cellular homeostasis by mediating the degradation and recycling of proteins, lipids, and other macromolecules. As such, it serves as a central hub for both proteostasis and lipostasis. This review outlines genetic and mechanistic parallels between rare lysosomal lipid storage diseases, such as Gaucher disease and Niemann–Pick disease, and more prevalent neurodegenerative diseases. We discuss how impaired lysosomal sphingolipid metabolism compromises lysosomal integrity, disrupts proteostasis, and contributes to neurodegeneration. Furthermore, we describe how age-related decline in lysosomal function may similarly drive neurodegeneration in the absence of overt genetic mutations. Taken together, this review highlights the lysosome as a central integrator of protein and lipid homeostasis and emphasizes the bidirectional relationship between lipostasis and proteostasis, whereby disruption of one adversely affects the other in the pathogenesis of multiple neurodegenerative diseases.

## 1. Introduction

Neurodegenerative diseases, including Alzheimer’s disease (AD), Parkinson’s disease (PD), and related disorders, represent a growing burden to aging societies. These disorders are defined by the progressive accumulation and spreading of misfolded proteins that disrupt cellular function and ultimately lead to neuronal loss and cognitive or motor decline [1,2]. A central driver of these pathogenic cascades is the breakdown of intracellular degradation systems, particularly the lysosomal degradation system, which is responsible for the clearance of aggregated proteins and damaged organelles. Its dysfunction not only leads to an accumulation of intracellular protein aggregates but may also facilitate the prion-like intercellular propagation of pathogenic species, hallmarks observed across many neurodegenerative diseases [3,4,5,6,7].

In addition to its well-recognized role in protein homeostasis (proteostasis), the lysosome is also a central regulator of lipid homeostasis (lipostasis) [8]. Neurons, given their high surface-area-to-volume ratio and extensive metabolic demands, are especially reliant on efficient lipid turnover for structural maintenance and metabolic support. Disruptions in the lipid metabolism, particularly in sphingolipid processing, give rise to lysosomal lipid storage disorders [9].

Recent findings indicate that protein and lipid degradation pathways converge within the lysosome, and that their dysfunction may act synergistically to drive neurodegeneration. This review explores the reciprocal relationship between lipid metabolism and proteostasis. Using genetic and molecular insights from rare monogenic lysosomal lipid storage disorders and common age-associated neurodegenerative diseases, we discuss how defects in lipid homeostasis can affect protein quality control and, conversely, how perturbations in proteostasis can affect lipid handling.

Taken together, these insights could offer a unified perspective for understanding these diseases.

## 2. The Role of Lysosomes in Protein Homeostasis

To maintain proteostasis and prevent the accumulation of misfolded or aggregation-prone proteins, cells rely on two proteolytic machineries: the proteasome and the lysosome. While both systems cooperate to ensure protein quality control, they operate through distinct mechanisms and target different classes of substrates. The ubiquitin–proteasome system (UPS) primarily handles the degradation of short-lived, regulatory, or misfolded proteins, accounting for the turnover of approximately 80–90% of intracellular proteins [10]. In contrast, the lysosome specializes in the clearance of bulkier and more complex substrates, including long-lived proteins, protein aggregates, and organelles [10].

Lysosomal degradation is supplied by three major input pathways: phagocytosis, endocytosis, and autophagy (Figure 1A). Phagocytosis and endocytosis mediate the uptake of extracellular material. While phagocytosis involves the engulfment of large particles such as pathogens or cellular debris, endocytosis is responsible for the internalization of soluble molecules and membrane proteins [11,12,13]. Following internalization, cargo is delivered to early endosomes (EEs), which function as sorting stations [12]. Material destined for recycling is returned to the plasma membrane via recycling endosomes (REs), whereas cargo targeted for degradation is sorted into intraluminal vesicles (ILVs) [14]. ILV formation begins in EEs and is mediated by the endosomal sorting complex required for transport (ESCRT), which recognizes ubiquitinated substrates and drives vesicle budding [11,14]. As EEs mature into late endosomes (LEs), marked by a Rab5-to-Rab7 switch, they continue to accumulate ILVs and develop into multivesicular bodies (MVBs), which eventually fuse with lysosomes to initiate degradation [11,14].

Autophagy is the major pathway for the removal of intracellular material. It involves the sequestration of cytoplasmic cargo within double-membraned autophagosomes, which either fuse directly with lysosomes or first merge with EEs or LEs to form amphisomes [15,16]. These hybrid organelles represent a point of convergence between autophagic and endocytic pathways [15,16], ensuring flexibility and efficiency in substrate delivery to lysosomes. The vesicular fusion events that mediate these transitions rely on shared molecular machinery, including SNAREs, Rab GTPases, and tethering complexes, which coordinate membrane docking and fusion.

Reflecting the diversity of cargo and the convergence of multiple input pathways, lysosomes comprise a heterogeneous population of vesicles that vary in composition, morphology, and subcellular localization [17]. To indicate their origin, these terminal degradative compartments are sometimes referred to as autolysosomes or endolysosomes. However, for the sake of clarity and simplicity, we refer to them collectively as lysosomes throughout this review.

While lysosomes were once viewed merely as cellular waste bags, they are now recognized as highly dynamic organelles involved in a broad range of processes. Beyond the degradation of damaged proteins and organelles, they function as central signaling platforms that sense nutrient and energy status, coordinate metabolic adaptations, and influence cell fate decisions [17]. They also maintain ion homeostasis and contribute to immune responses and stress adaptation. Additionally, lysosomes interact with other organelles and facilitate their communication. As such, lysosomes play a key role in maintaining cellular homeostasis, and their dysfunction can lead to a variety of defects.

## 3. The Role of the Lysosomal Degradation System in Neurodegenerative Diseases

Neurodegenerative diseases, such as AD and PD, are characterized by the progressive loss of specific neuronal populations, resulting in cognitive decline, motor impairment, and other neurological deficits [18]. Despite their clinical differences, these disorders share key pathological hallmarks, such as the accumulation and aggregation of disease-specific proteins, for example tau (MAPT) in AD and other tauopathies and α-synuclein (SNCA) in PD and other synucleinopathies [19,20]. These intracellular inclusions disrupt essential cellular processes and contribute directly to neuronal toxicity [21].

A growing body of pathological and genetic evidence implicates impaired lysosomal degradation in the development and progression of neurodegenerative diseases (Figure 1B). Postmortem brain tissue from affected individuals consistently reveals morphological abnormalities in intracellular vesicles, including enlarged lysosomes, accumulated autophagosomes, and vesicles filled with incompletely degraded substrates, indicative of defective cargo turnover [22]. Similar phenotypes have been observed in animal and cellular models, where the accumulation of dysfunctional autophagic and lysosomal vesicles strongly correlates with neurotoxicity [23,24]. These alterations are often observed prior to overt neuronal loss, suggesting a role in disease initiation [23,24].

Genetic studies provide further support for a central role of the lysosome in neurodegeneration. For example, in familial AD, mutations in presenilin 1 (PSEN1) impair the assembly of the v-ATPase complex, thereby reducing lysosomal acidification and degradative capacity [25]. In familial PD, mutations in VPS35, a component of the retromer complex involved in endosomal sorting, compromise the delivery of hydrolases to the lysosome [26]. Moreover, leucine-rich repeat kinase 2 (LRRK2), a kinase mutated in certain familial forms of PD, has been implicated, among other roles, in lysosomal membrane repair through the modulation of the ESCRT pathway in macrophages and astrocytes [27,28]. While these mutations have pleiotropic effects, their impact on lysosomal function is a shared feature that likely contributes to disease pathogenesis by impairing the neuronal capacity to process and clear misfolded proteins, leading to their accumulation.

Importantly, impaired lysosomal degradation is not only associated with intracellular protein accumulation but also plays a pivotal role in the intercellular propagation of pathological proteins. Misfolded proteins can be secreted via exosomes or other vesicle-associated pathways, or as free proteins, and subsequently taken up by neighboring or synaptically connected cells [29]. The inhibition of lysosomal clearance can enhance both the secretion and intercellular spreading of pathological proteins [30,31]. Upon internalization, these pathological proteins were shown to accumulate within endolysosomes, where they can destabilize and ultimately rupture their membranes [32,33,34,35,36,37,38,39,40]. This rupture seems to permit the release of fibrillar seeds into the cytosol, where they act as templates to recruit and convert native proteins into aggregates, thereby initiating and amplifying pathology [32,33,34,35,36,37,38,39,40]. Notably, endolysosomal escape is emerging as a rate-limiting step in prion-like propagation and can be accelerated either by direct damage of the endolysosomal membranes or by impairing the activity of membrane repair pathways, such as ESCRT machinery [32,34,35].

Taken together, the lysosomal degradation system safeguards neuronal health by enabling the efficient clearance of misfolded proteins and damaged organelles. When this system is compromised, it not only leads to the intracellular accumulation of protein aggregates but also promotes their intercellular transmission and propagation, thereby driving the progression of neurodegenerative diseases.

## 4. The Role of Lysosomes in Lipid Homeostasis

In addition to its well-established role in protein turnover, the lysosome serves as a central hub for lipid degradation. Lipid substrates are delivered to the lysosome from both external sources, such as lipoproteins internalized via endocytosis, and internal sources, including organelles and lipid droplets targeted by autophagy [41]. In autophagy, lipids are sequestered within the inner membrane of autophagosomes, whereas endocytosed lipids are initially incorporated into the limiting membrane of EE and subsequently sorted into ILVs [41]. This spatial reorganization is crucial, as the limiting membrane is protected by a glycoprotein-rich glycocalyx that faces the lysosomal lumen, rendering it inaccessible to lysosomal hydrolases [42]. Only once sorted into ILVs can lipid substrates be processed by the enzymatic machinery of the lysosome.

Within the acidic environment of the lysosomal lumen, lipid degradation is carried out by a diverse repertoire of acid hydrolases. Glycerophospholipids are hydrolyzed by phospholipase A (PLA) and phospholipase B (PLB) enzymes, such as PLA2G15 and PLBD2, yielding lysophospholipids, glycerophosphodiesters (GPDs), and free fatty acids [43,44]. These products are exported from the lysosome by specific transporters, including SPNS1 for lysophospholipids and CLN3 for GPDs [45,46].

Neutral lipids such as cholesteryl esters and triacylglycerols are degraded by lysosomal acid lipase (LAL), encoded by the *LIPA* gene, generating free fatty acids, cholesterol, and glycerol [47]. Cholesterol is subsequently exported via the NPC1 and NPC2 transporters [48,49,50].

While many lipid classes can be directly degraded by hydrolases, sphingolipids require additional cofactors due to their complex structures and strong membrane association [51]. These lipids consist of a sphingosine backbone, a long-chain amino alcohol, which is linked to a fatty acid via an amide bond [52]. Based on their polar head groups, sphingolipids are classified into several subtypes, including ceramides, sphingomyelins, and glycosphingolipids (GSLs). GSLs are further divided into cerebrosides, sulfatides, globosides, and gangliosides.

The catabolism of sphingolipids is a tightly controlled, multi-step process in which each enzymatic step is mediated by a specific lysosomal hydrolase [9] (Figure 2). For example, glucocerebrosidase degrades glucosylceramide, while sphingomyelinase hydrolyzes sphingomyelin [9]. However, efficient breakdown also requires lipid activator proteins such as saposins, which extract sphingolipids from ILV membranes and present them to their respective hydrolases [53]. Additionally, the ILV membrane itself plays an active role in facilitating degradation: it is enriched in bis(monoacylglycero)phosphate (BMP), an anionic phospholipid that remains negatively charged at lysosomal pH and enhances the binding and activity of cationic lysosomal enzymes at the membrane–water interface [54]. The end products of sphingolipid catabolism include ceramide, sphingosine, fatty acids, and polar head groups such as monosaccharides and sulfate [55]. These products are recycled from the lysosome and serve as substrates for energy production or as building blocks in biosynthetic pathways. The brain, as one of the most lipid-rich organs, contains particularly high levels of phospholipids, including sphingolipids and glycerophospholipids, as well as cholesterol, and therefore relies heavily on lipid metabolism to maintain cellular function [56].

Lipids serve several roles in the nervous system. They act as an energy source, especially under conditions of metabolic stress, and contribute structurally to the extensive membrane networks of neurons [57]. Sphingolipids and cholesterol-rich membrane microdomains facilitate the association of specific signaling molecules and receptors [58]. In myelinated axons, lipids have both structural and functional roles. Myelin is rich in sphingolipids and saturated very long-chain fatty acids, which tightly pack together to increase membrane rigidity, essential for creating a low-permeability barrier [56]. This property is critical for axonal insulation and the rapid propagation of action potentials via saltatory conduction [56,59]. Beyond their structural roles, certain sphingolipids, such as ceramide and sphingosine-1-phosphate (S1P), also function as bioactive signaling molecules, regulating crucial cellular processes including development, differentiation, apoptosis, and inflammation [60].

Overall, lysosomal lipid turnover, particularly of sphingolipids, is indispensable not only for energy and biosynthetic needs but also for maintaining membrane composition and integrity, organelle function, and signaling, especially in neurons.

## 5. Failure of Lipid Degradation Leads to Lipidosis

Defective lysosomal lipid degradation—whether due to mutations in hydrolases, activator proteins, or membrane-associated factors—leads to the accumulation of undegraded lipids, a defining hallmark of lysosomal lipid storage disorders (lipidoses) [9].

These disorders primarily include sphingolipidoses, Niemann–Pick type C (NPC), and Wolman disease, as well as its milder form, cholesteryl ester storage disease (CESD). Both Wolman disease (~1% residual activity) and CESD (~10% residual activity) are caused by mutations in *LIPA*, causing LAL deficiency [61]. In NPC, mutations in NPC1 or NPC2 impair the export of unesterified cholesterol and other lipids from lysosomes, leading to their accumulation [61].

The sphingolipidoses represent a major subclass of lipidoses, reflecting the enzymatic complexity required for sphingolipid catabolism. These disorders are caused by mutations in enzymes responsible for degrading specific sphingolipid species, resulting in substrate accumulation of their respective substrates [62]. Prominent examples include Gaucher disease, Tay–Sachs disease, and Niemann–Pick disease types A and B. An overview is provided in Figure 2.

The exact molecular mechanisms by which the intra-lysosomal accumulation of undegraded lipids leads to cellular and tissue dysfunction remain poorly understood, particularly given the broad consequences of lysosomal impairment. While once considered static “garbage disposal” units, lysosomes are now recognized as dynamic organelles involved in a broad array of cellular functions, including the metabolic signaling, and regulation of gene expression, immune responses, and membrane repair [17]. Their ability to sense and adapt to changing metabolic demands and to physically interact with other organelles positions lysosomes as central regulators of cellular homeostasis. Given the critical roles of lipids in neuron function, it is not surprising that many lipidoses affect the nervous system, often presenting with neurological symptoms such as motor deficits, cognitive impairment, seizures, and progressive neurodegeneration.

Notably, several pathological features observed in lipidoses, such as impaired autophagy, endolysosomal membrane rupture, and intracellular protein aggregation, mirror those found in more common neurodegenerative diseases, including PD and AD [63]. This overlap suggests that lysosomal lipid dysregulation may secondarily impair proteostasis. Understanding the mechanistic cross-talk between lipostasis and proteostasis within the lysosome may reveal shared pathogenic pathways that drive both rare lipidoses and common neurodegenerative disorders.

## 6. Commonalities Between Sphingolipidosis and Neurodegenerative Diseases

Emerging evidence reveals genetic and molecular relationships between sphingolipidosis and neurodegenerative diseases. Several lysosomal genes originally identified in the context of rare sphingolipidoses have since been implicated in major neurodegenerative diseases. To better understand the convergence between lysosomal lipid storage disorders and neurodegenerative diseases, it is instructive to examine individual genes that lie at this intersection. In the following sections, we highlight key examples focusing on how their dysfunction impacts both lipid metabolism and proteostasis, and how these disruptions contribute to neurodegenerative processes. An overview can be found in Table 1.

### 6.1. GBA1

Homozygous mutations in *GBA1*, which encodes the lysosomal enzyme β-glucocerebrosidase (GCase), cause Gaucher’s disease, characterized by reduced GCase activity and the accumulation of glucosylceramide and glucosylsphingosine [64]. Beyond Gaucher’s, *GBA1* mutations are the most common genetic risk factor for PD [65]. Some studies have shown that GCase deficiency alters lysosomal lipid composition, disrupts chaperone-mediated autophagy, and promotes α-synuclein aggregation and enhanced cell-to-cell transmission [66,67,68,69]. These changes can also destabilize the lysosomal membrane, increasing the risk of rupture [67] Notably, GCase activity also declines with age in the absence of mutations, suggesting that lysosomal vulnerability may contribute broadly to sporadic PD [70].

### 6.2. NPC1

Mutations in *NPC1* cause NPC, a lysosomal storage disorder characterized by impaired cholesterol export [71]. The massive buildup of cholesterol seems to jam the autophagic and endolysosomal systems and impair intracellular trafficking, thereby compromising the efficient delivery of newly synthesized hydrolases to the lysosome [71,72,73]. As a result, the degradation capacity for both lipids and proteins becomes even further compromised. Hence, in addition to cholesterol, this also leads to the accumulation of, e.g., glycosphingolipids. Moreover, NPC1-deficient lysosomes exhibit more fragile membranes, as shown by the increased recruitment of galectin-3 and ESCRT machinery in response to the application of lysosome-destabilizing agents [74], suggesting a higher propensity for membrane rupture. As discussed above, these leaky lysosomes may facilitate the intra- and inter-cellular propagation of amyloid-like proteins, accelerating their aggregation [75,76,77]. Ultimately, the leakage of lysosomal hydrolases into the cytosol can trigger apoptotic pathways, leading to cell death and tissue damage [78]. The potential involvement of *NPC1* mutations in PD remains a subject of debate. Some studies have reported individuals with PD or parkinsonian syndromes who are heterozygous carriers of *NPC1* mutations [79]. However, other investigations have not found compelling evidence that *NPC1* variants play a significant role in synucleinopathies [80,81], indicating that the link, if present, may be limited or context dependent.

### 6.3. SMPD1

In contrast to NPC, NPA and NPB are caused by mutations in *SMPD1*, which encodes acid sphingomyelinase (ASM), the enzyme responsible for degrading sphingomyelin into ceramide and phosphorylcholine (Figure 2). A deficiency of ASM leads to sphingomyelin accumulation, classifying these disorders as classic sphingolipidoses. Homozygous *SMPD1* mutations are linked to NPA, the severe neuronopathic form, which presents in infancy with rapid neurodegeneration [82]. Heterozygous variants of *SMPD1* are associated with an increased risk of PD [83,84], suggesting a link between impaired sphingolipid degradation and PD.

### 6.4. GALC

Mutations in *GALC*, which encodes galactocerebrosidase, cause Krabbe disease, a lysosomal lipid storage disorder characterized by the accumulation of psychosine (galactosylsphingosine). In the absence of functional galactocerebrosidase, psychosine builds up in oligodendrocytes and neurons, ultimately leading to severe demyelination and neurodegeneration [85]. Psychosine inserts into membranes and interferes with their structural integrity and signaling function [86]. Notably, the effects of psychosine extend beyond lipostasis disruption. α-Synuclein aggregates isolated from postmortem brains of Krabbe disease patients exhibit prion-like properties, suggesting a potential link between *GALC* dysfunction and α-synuclein propagation [87]. Indeed, psychosine was shown to directly promote α-synuclein misfolding and accumulation [88,89]. Furthermore, a genome-wide association and rare variant analyses identified *GALC* as a genetic risk factor for PD [90].

### 6.5. GRN

Mutations in *GRN*, encoding progranulin (PGRN), represent another strong connection between lipidoses and neurodegenerative diseases. The complete loss of PGRN function, typically due to homozygous *GRN* mutations, causes neuronal ceroid lipofuscinosis (NCL, also known as Batten disease), a classical lysosomal storage disorder marked by the accumulation of ceroid lipofuscin [91]. Haploinsufficiency, often resulting from heterozygous *GRN* mutations, leads to *GRN*-associated frontotemporal dementia (GRN-FTD), which accounts for 10–15% of all FTD cases and is characterized by lipofuscinosis, microgliosis, TDP-43 pathology, and cortical neuronal loss [92]. Notably, *GRN* variants have also been identified in patients with AD and PD, indicating a broader relevance for PGRN in more common neurodegenerative diseases [93].

PGRN seems to play a key role in lysosomal lipid metabolism. It interacts with several lysosomal proteins involved in sphingolipid degradation, including GCase and prosaposin, and supports the proper localization and activity of GCase [94]. PGRN deficiency reduces GCase function, leading to the accumulation of glucosylceramide, a lipid implicated in Gaucher disease and an enhancer of tau and α-synuclein aggregation, pathological features of AD and PD [95]. Furthermore, PGRN deficiency is associated with reduced levels of BMP, further compromising GCase activity [92].

However, contrary to the view that PGRN deficiency causes neurotoxicity through sphingolipid accumulation, recent lipidomic and proteomic analyses of GRN-FTD brain tissue revealed a striking loss of myelin-enriched sphingolipids (sulfatide, galactosylceramide, sphingomyelin) in frontal white matter [96]. Notably, GALC activity, which is responsible for breaking down galactosylceramide and sulfatide, was selectively increased in GRN-FTD, suggesting a hyperactive myelin lipid catabolism [96]. This may represent a compensatory response to lysosomal dysfunction and may contribute to myelin breakdown, gliosis, and neurodegeneration in GRN-FTD.

Taken together, these findings suggest that PGRN deficiency may lead to a lysosomal lipid imbalance not only by impairing turnover but also by accelerating lipid degradation. This imbalance may drive protein misfolding and promote the accumulation of several disease-associated proteins and underscores the PGRN–GCase–lipid metabolism axis as a promising therapeutic target for multiple neurodegenerative diseases.

## 7. Genetic Screens Reveal Regulators of Lipid and Protein Homeostasis

In addition to the disease-associated genes discussed in the previous section, recent unbiased genetic screens have uncovered new genes and mechanisms involved in the cross-talk of lipostasis and proteostasis.

A genome-wide CRISPR interference screen identified multiple genes whose knock-down promoted protein aggregation, as measured by increased ProteoStat staining [97]. These included components of established proteostasis pathways, such as ribosomal subunits, proteasome components, v-ATPase subunits, and chaperones, as well as genes involved in lipid uptake and lysosomal sphingolipid metabolism (e.g., LDLR, GNPTAB, IGF2R, LYSET, and PSAP). Notably, aggregates were found to accumulate predominantly in lysosomes, and lipidomic analysis revealed a strong correlation between aggregate burden and elevated levels of sphingomyelins and cholesteryl esters.

In line with this, a genome-wide RNAi screen in a *C. elegans* tauopathy model identified several genes involved in sphingolipid metabolism as critical regulators of endolysosomal integrity [98]. Follow-up studies showed that impairing sphingolipid metabolism increased membrane rigidity, rendering endolysosomal vesicles more prone to rupture [99]. Given the high sphingolipid content of lysosomal membranes, they are likely particularly susceptible to sphingolipid perturbations [100]. Moreover, fibrillar tau aggregates further exacerbated this rigidity, establishing a pathological feedback loop in which sphingolipid imbalance and tau pathology amplify one another [99]. Increased membrane rigidity promoted seeded tau aggregation, likely by facilitating the escape of tau seeds from compromised endolysosomal compartments into the cytosol [99]. Together, these studies provide compelling evidence that lipid homeostasis and proteostasis are tightly interconnected, and that disruption of the sphingolipid metabolism can destabilize lysosomes, thereby exacerbating protein aggregation.

## 8. Dose-Dependent Effects and the Aging Brain

Importantly, many of the disease-associated genes exert dose-dependent effects: while biallelic mutations cause classical early-onset lipidoses, heterozygous mutations increase the risk for late-onset neurodegenerative diseases [64]. These findings suggest that partial impairment of lysosomal lipid degradation, below the threshold of rare lipid storage disease, may contribute to the pathogenesis of common neurodegenerative diseases later in life. Following this logic, one could envision that even subtle changes in enzyme activity could gradually impair lysosomal lipid turnover. These effects are likely context-dependent, influenced by factors such as cell type or the presence of additional genetic or environmental risk modifiers, and may therefore fall below the detection threshold of genome-wide association studies. Over time, this persistent imbalance could eventually trigger neurodegenerative processes in later life. Indeed, growing evidence indicates that lipid dyshomeostasis occurs in the aging brain and in sporadic AD and PD [101,102].

Aging is the strongest risk factor for most neurodegenerative diseases, yet the cellular mechanisms underlying this vulnerability are complex and multifactorial. One key contributor is the progressive decline in lysosomal enzyme activity leading to the accumulation of lipid and protein substrates, which does not only impair autophagic flux [9,103]. Another increasingly recognized consequence of this decrease is the dysregulation of sphingolipid metabolism, marked by elevated levels of bioactive lipid species such as ceramide, sphingosine, and ceramide-1-phosphate [104]. Elevated ceramide concentrations can compromise the integrity of lysosomal membranes and trigger the release of catabolic enzymes and apoptotic factors into the cytosol, promoting cellular stress and neurotoxicity [103,105]. Importantly, these lipid perturbations are not restricted to the aging process itself. Similar changes have been consistently observed in postmortem brain tissue from patients with sporadic AD [57,106,107]. These findings suggest that lipid dyshomeostasis is not merely a downstream consequence of neurodegeneration, but a potential upstream driver, initiating or amplifying lysosomal dysfunction and impairing the clearance of toxic protein aggregates.

Additionally, lysosomal dysfunction has been implicated in the induction and maintenance of cellular senescence. Senescent cells release cytokines, proteases, and lipotoxic factors as part of the senescence-associated secretory phenotype, contributing to chronic inflammation and tissue damage [108]. This pro-inflammatory environment further impairs proteostasis and drives neurodegenerative diseases.

Taken together, these observations indicate that even in the absence of genetic mutations, age-related deterioration of the lysosomal lipid metabolism can undermine proteostasis and contribute to neuronal vulnerability. This highlights the importance of considering lipid–protein interactions within the lysosome as a key axis of dysfunction in both genetic and sporadic forms of neurodegeneration.

## 9. Conclusions

Collectively, the evidence presented here highlights the central role of the lysosomal system in both lipid and protein turnover. Sphingolipid catabolism, in particular, emerges as a critical regulator of neuronal homeostasis. Disruption of this finely balanced system, whether through genetic mutations, age-related decline, or environmental stressors, can compromise lysosomal function, promote the accumulation of toxic substrates, and initiate neurodegeneration. The findings summarized in this review underscore the intimate interplay between proteostasis and lipostasis within the lysosome. Understanding the molecular mechanisms that bridge these two processes will be key to developing targeted therapies for neurodegenerative diseases.

## 10. Outlook

Emerging evidence suggests that disruptions in either lipid or protein degradation can initiate a self-perpetuating pathogenic cycle. For instance, sphingolipid accumulation can impair autophagic flux, thereby promoting the aggregation of misfolded proteins. Conversely, the buildup of protein aggregates may interfere with endolysosomal trafficking and compromise the delivery of lysosomal enzymes, disrupting both protein and lipid degradation. In this manner, lipid and protein pathologies may reinforce one another, escalating cellular stress and contributing to progressive neurodegeneration. A deeper understanding of this bidirectional relationship between lipid metabolism and proteostasis is essential and may offer new avenues for therapeutic intervention. This reciprocal interference may also help explain the frequent coexistence of multiple aggregation-prone proteins in neurodegenerative diseases. The misfolding of one key protein could impair lysosomal function, leading to the secondary toxic accumulation of lipids and promoting the aggregation of additional disease-associated proteins such as α-synuclein or TDP-43, thereby driving comorbid proteinopathies.

Aging introduces an additional layer of complexity. Subtle, progressive alterations in sphingolipid metabolism may occur long before clinical symptoms emerge, suggesting a potential window for early intervention. Although substantial progress has been made in understanding how lipid and protein homeostasis intersects within the lysosome, several questions remain unresolved.

To address these challenges, future research should aim to:Leverage lipidomics for biomarker discovery: High-resolution lipid profiling holds promise for identifying early diagnostic and prognostic markers of neurodegenerative diseases. Achieving this goal will require the standardization of lipidomics methodologies across laboratories and platforms.Investigate the role of lipid droplets in neural cells: Accumulation of lipid droplets in neurons and glia has been linked to neurodegeneration [109]. Understanding the dynamics and function of lipid droplets could uncover new therapeutic targets.Evaluate the therapeutic potential of fatty acid substitution, including polyunsaturated fatty acids: Dietary supplementation with, e.g., polyunsaturated fatty acids may help rebalance lipid metabolism and reduce neuroinflammation, particularly in aging and early disease stages.Understanding disease specificity: Several lysosomal genes have been linked to distinct neurodegenerative diseases (e.g., *GBA1* to PD, *GRN* to FTD), but the basis for this specificity remains unclear. Future work should explore how factors such as particular lipid metabolites, cell type, or potential regional differences in brain lipid composition might influence which disease phenotype emerges in response to a given lysosomal defect.Clarify the interplay between genetic and environmental factors in lipid–protein homeostasis: Determining how risk genes and environmental stressors influence lipostasis and proteostasis in neurons will be essential for identifying susceptible populations and designing preventive interventions.

The integration of lipidomics, proteomics, and functional genomics across experimental models and human tissues will be key to unraveling these complex relationships. Ultimately, therapies aimed at stabilizing lysosomal function or restoring sphingolipid balance may offer broad benefits across a spectrum of neurodegenerative diseases.

## Figures and Tables

**Figure 1 cells-14-00845-f001:**
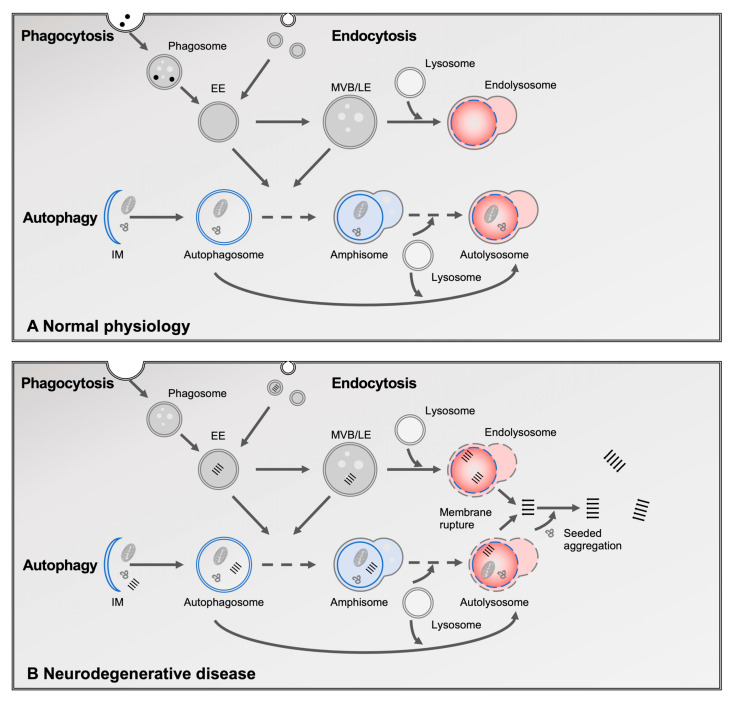
Overview of the autophagy–lysosomal system and its dysfunction in neurodegenerative diseases. (**A**) The lysosome serves as the terminal degradation hub for three major intracellular trafficking pathways: autophagy, endocytosis, and phagocytosis. In autophagy, cytosolic components—including protein aggregates, damaged organelles, and lipids—are enclosed starting with the isolating membrane (IM) to form a double-membraned autophagosome. Autophagosomes subsequently fuse with endosomes to form amphisomes, and then with lysosomes to create autolysosomes, where degradation occurs. Endocytosis delivers extracellular materials, including lipids and receptors, to early endosomes (EE), which mature into late endosomes (LE) and multivesicular bodies (MVBs) before fusing with lysosomes. Phagocytosis, primarily in microglia and other immune cells, engulfs large extracellular particles such as apoptotic cells and debris into phagosomes, which also fuse with lysosomes for degradation. (**B**) In neurodegenerative diseases, this system is impaired at multiple levels. A key feature is lysosomal membrane rupture, which allows aggregated proteins to escape into the cytosol, promoting seeded aggregation.

**Figure 2 cells-14-00845-f002:**
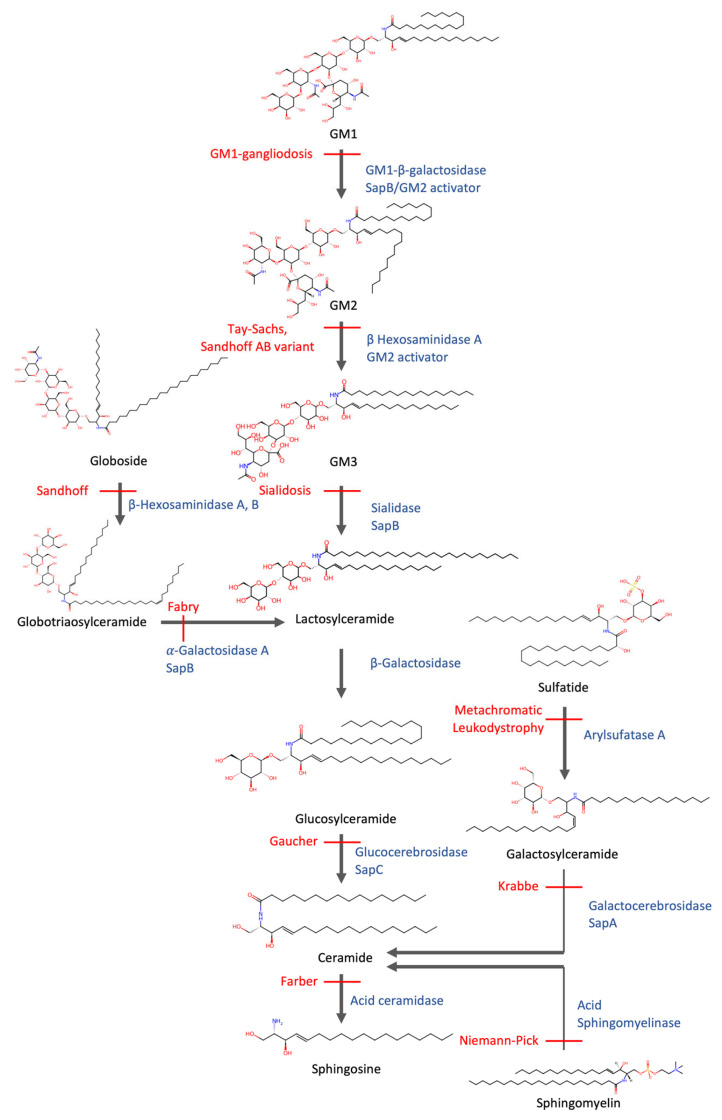
Degradation of sphingolipids and associated sphingolipidoses. Sphingolipids are catabolized in a sequential, enzyme-dependent process within lysosomes. Enzyme deficiencies (blue) lead to substrate accumulation and define distinct lysosomal lipid storage disorders (red), including Tay–Sachs, Gaucher, Krabbe, and Niemann–Pick diseases. This pathway highlights how disruption at nearly any step can result in a specific sphingolipidosis. Chemical structures provided by ChemSpider, 2025, https://www.chemspider.com/Chemical-Structure (accessed on 15 May 2025).

**Table 1 cells-14-00845-t001:** Overview of the pathological and mechanistic overlap between sphingolipidoses and neurodegenerative diseases. ↑: indicates increased risk.

Gene	Lipid Storage Disorder	Lipid Accumulation or Imbalance	Neurodegeneration-Linked Pathology	Shared Mechanisms
*GBA1*	Gaucher disease	glucosylceramide, glucosylsphingosine	↑ PD risk; α-synuclein aggregation and transmission	autophagy impairment, lysosomal rupture, proteostasis failure
*NPC1*	Niemann–Pick type C	unesterified cholesterol, glycosphingolipids	lysosomal rupture; debated PD association	autophagy impairment, lysosomal enzyme mistrafficking, membrane rupture
*SMPD1*	Niemann–Pick type A/B	sphingomyelin	↑ PD risk (variants)	autophagy impairment, lysosomal stress
*GALC*	Krabbe disease	psychosine (galactosyl-sphingosine)	↑ PD risk; aggregation and prion-like propagation of α-synuclein	lysosomal stress and membrane destabilization
*GRN*	NCL and *GRN*-FTD	glucosylceramide; loss of myelin lipids (e.g., sulfatide)	TDP-43 aggregation, cortical neuronal loss; ↑ AD and PD risk	disrupted GCase activity (see *GBA1*), myelin loss

## Data Availability

No new data were created or analyzed in this study.

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
