# Peer review of "Broken Balance: Emerging Cross-Talk Between Proteostasis and Lipostasis in Neurodegenerative Diseases"

_cells, 2025, doi:10.3390/cells14110845_

Round 1
Reviewer 1 Report
Comments and Suggestions for Authors
This is an educational and comprehensive review article about a timely and interesting topic.
The concerns are minor:
- The focus on sphingolipids and lysosomal lipid degradation could maybe be more explicit and reflected in the title, but maybe it is ok as it is
- The authors could address more (even though we don't know) why there is a certain specificity: GBA is linked to PD, NPC to NPD, PGRN to FTD... Yes, there is co-pathology, but still there is specificity that should be addressed at least by speculating why.
- Alzheimer's disease seems to be less relevant here, and that could also be addressed.
- Other minor concerns can be found in the attached annotated pdf.

Author Response
Reviewer 1
This is an educational and comprehensive review article about a timely and interesting topic.
We thank the reviewer for the positive feedback. We appreciate the constructive suggestions, which
have helped us further clarify the manuscript.
The concerns are minor:
The focus on sphingolipids and lysosomal lipid degradation could maybe be more explicit and
reflected in the title, but maybe it is ok as it is.
We would prefer to retain the original wording to better reflect the dual focus on lipid and protein
metabolism within the lysosome.
The authors could address more (even though we don't know) why there is a certain specificity: GBA
is linked to PD, NPC to NPD, PGRN to FTD... Yes, there is co-pathology, but still there is specificity
that should be addressed at least by speculating why.
This is a very interesting aspect. We believe that this is a key area for future investigation. Therefore,
we have added a new point in the Outlook section 10 that also speculates on potential contributing
factors (lines 477-482): “Understanding disease specificity: Several lysosomal genes have been linked
to distinct neurodegenerative diseases (e.g., GBA1 to PD, GRN to FTD), but the basis for this
specificity remains unclear. Future work should explore how factors such as particular lipid
metabolites, cell type, or potential regional differences in brain lipid composition, composition might
influence which disease phenotype emerges in response to a given lysosomal defect.”
Alzheimer's disease seems to be less relevant here, and that could also be addressed.
We agree with the reviewer that, from a genetic standpoint, fewer mutations in sphingolipid
metabolism genes have been associated with AD compared to PD or FTD. However, as we discuss in
the manuscript, there is growing evidence of sphingolipid dysregulation in AD brain tissue. While
these alterations may be more subtle and potentially influenced by other risk factors, they may still
contribute to disease susceptibility or progression. To acknowledge this, we have added the following
speculative statement to section 8 (lines 402-406): “Following this logic, one could envision that even
subtle changes in enzyme activity could gradually impair lysosomal lipid turnover. These effects are
likely context-dependent, influenced by factors such as cell type, or the presence of additional genetic
or environmental risk modifiers, and may therefore fall below the detection threshold of genome-wide
association studies.”
Other minor concerns can be found in the attached annotated pdf.
We thank the reviewer for their detailed feedback. Minor textual revisions have been made
accordingly throughout the manuscript.

Reviewer 2 Report
Comments and Suggestions for Authors
The manuscript by Jessica Tittelmeier and Carmen Nussbaum-Krammer provides a clear and well-structured review of the general roles of lysosomes in cellular and lipid homeostasis. The topic is timely and scientifically relevant, particularly given the growing recognition of lysosomal dysfunction in lipid storage disorders and the role of sphingolipidosis in the pathogenesis of neurodegenerative diseases.
The review addresses an important and current area of research. The manuscript is generally well organized and the arguments are clearly presented.
Minor Comments:
- In Section 2: “The role of lysosomes in protein homeostasis”, the authors could expand on the notion of lysosomes as dynamic organelles involved in a broad range of cellular functions beyond degradation, such as metabolic regulation, gene expression, and stress responses. This would help counteract the outdated perception of lysosomes merely as “cellular cleaning stations.”
- It would be beneficial to include a summary table synthesizing the key points discussed in Section 6: “Commonalities between sphingolipidosis and neurodegenerative diseases”, particularly highlighting the pathological correlations between these conditions
- In Section 8: “Dose-dependent effects and the aging brain”, a brief discussion on the role of lysosomal dysfunction in cellular senescence would strengthen the section and provide a more comprehensive view of lysosome-related aging processes.
Author Response
Reviewer 2
The manuscript by Jessica Tittelmeier and Carmen Nussbaum-Krammer provides a clear and wellstructured
review of the general roles of lysosomes in cellular and lipid homeostasis. The topic is
timely and scientifically relevant, particularly given the growing recognition of lysosomal dysfunction
in lipid storage disorders and the role of sphingolipidosis in the pathogenesis of neurodegenerative
diseases.
The review addresses an important and current area of research. The manuscript is generally well
organized and the arguments are clearly presented.
We thank the reviewer for their positive assessment of our manuscript and for recognizing the
relevance and timeliness of the topic. We are pleased that the overall structure and clarity of our
arguments were well received. We appreciate the reviewer’s thoughtful engagement and constructive
feedback, which have helped us improve the manuscript further.
Minor Comments:
- In Section 2: “The role of lysosomes in protein homeostasis”, the authors could expand on the notion
of lysosomes as dynamic organelles involved in a broad range of cellular functions beyond
degradation, such as metabolic regulation, gene expression, and stress responses. This would help
counteract the outdated perception of lysosomes merely as “cellular cleaning stations.”
We have added the following paragraph to section 2 (lines 89-96): “While lysosomes were once
viewed merely as cellular waste bags, they are now recognized as highly dynamic organelles involved
in a broad range of processes. Beyond the degradation of damaged proteins and organelles, they
function as central signaling platforms that sense nutrient and energy status, coordinate metabolic
adaptations, and influence cell fate decisions [17]. They also maintain ion homeostasis and contribute
to immune responses and stress adaptation. Additionally, lysosomes interact with other organelles
and facilitate their communication. As such, lysosomes play a key role in maintaining cellular
homeostasis, and their dysfunction can lead to a variety of defects.”
- It would be beneficial to include a summary table synthesizing the key points discussed in Section 6:
“Commonalities between sphingolipidosis and neurodegenerative diseases”, particularly highlighting
the pathological correlations between these conditions
We agree with the reviewer and have added a summary table to the end of section 6 (lines 368-369).
- In Section 8: “Dose-dependent effects and the aging brain”, a brief discussion on the role of
lysosomal dysfunction in cellular senescence would strengthen the section and provide a more
comprehensive view of lysosome-related aging processes.
We thank the reviewer for this suggestion. We have added the following paragraph in section 8 (lines
423-427): “Additionally, lysosomal dysfunction has been implicated in the induction of cellular
senescence. Senescent cells release cytokines, proteases and lipotoxic factors as part of the senescenceassociated
secretory phenotype, contributing to chronic inflammation and tissue damage [108]. This
pro-inflammatory environment further impairs proteostasis and drives neurodegenerative diseases."

Reviewer 3 Report
Comments and Suggestions for Authors
See attached file.

Author Response
Reviewer 3
The review paper by Tittelmeier and Nussbaum-Krammer reports on an emerging feature in
neurodegenerative diseases, i.e. the interplay between lipid and protein lysosomal metabolism and its
impairment in the pathogenesis of several neurodegenerative diseases. Lipid homeostasis is generally
given less attention than protein homeostasis in these diseases and the present review updates and
points out the many aspects connected with dysfunction of sphingolipid processing and its adverse
effects on protein homeostasis and neurodegeneration. The review is well written and comprehensive,
and I strongly support its publication in Cells. I only have a few minor points for attention of the
Authors before final acceptance.
We thank the reviewer for their thoughtful and supportive comments. We are pleased that the
reviewer found the review to be comprehensive, well written, and timely in addressing the interplay
between lipid and protein homeostasis in neurodegenerative diseases. We appreciate the recognition
of our effort to highlight the underexplored role of sphingolipid metabolism in this context. We have
carefully addressed the reviewer’s minor comments below.
1. Line 143: the term “pathology” is repeated twice.
Corrected
2. Page 5-6 and Figure 2: The section describing lipid catabolism would benefit from including a
scheme with the chemical structures of the main players in sphingolipids pathways. This will
help readers with biochemical background to appreciate the enzymatic steps and associated
diseases highlighted in Figure 2.
We thank the reviewer for this insight. We have added the chemical structures into Figure 2.
3. Line 310: the term “synuclein” at beginning of the sentence should be with a capital initial.
Corrected
4. Line 410, Conclusions section: It is not clear whether the sentence “Recent findings …” refers
to the present review or other papers. Please re-state more clearly
We apologize for the vague conclusion, we have adjusted the statement as follows (line 440):
“The findings summarized in this review underscore the intimate interplay between
proteostasis and lipostasis within the lysosome”.
